# Don't Read the Comments: Examining Social Media Discourse on Trans Athletes

**Zein Murib**

Department of Political Science, Fordham University, New York, NY 10023, USA; zmurib@fordham.edu

**Abstract:** How are transgender athletes understood in popular discourse? This paper adapts and merges Glaser and Strauss' 1967 Grounded Theory Method with computerized Automated Text Analysis to provide clarity on large-n datasets comprised of social media posts made about transgender athletes. After outlining the procedures of this new approach to social media data, I present findings from a study conducted on comments made in response to YouTube videos reporting transgender athletes. A total of 60,000 comments made on three YouTube videos were scraped for the analysis, which proceeded in two steps. The first was an iterative, grounded analysis of the top 500 "liked" comments to gain insight into the trends that emerged. Automated Text Analysis was then used to explore latent connections amongst the 60,000 comments. This descriptive analysis of thousands of datapoints revealed three dominant ways that people talk about transgender athletes: an attachment to biology as determinative of athletic abilities, a racialized understanding of who constitutes a proper "girl", and perceptions of sex-segregated sports as the sole way to ensure fairness in athletic opportunities. The paper concludes by drawing out the implications of this research for how scholars understand the obstacles facing transgender political mobilizations, presents strategies for addressing these roadblocks, and underscores the importance of descriptive studies of discourse in political science research concerned with marginalization and inequality.

**Keywords:** gender; transgender; athletics; fairness; biology; rights; politics

## 1. Introduction

In Spring 2017, Andraya Yearwood ran in her first track meet as a member of her high school girls' track team. "I felt very liberated, like a weight had been lifted off my shoulders", she said to Jezebel, an online feminist news site. "I was finally able to compete as who I knew that I was" (Wang 2021). The following season, Yearwood won the gold medal in the 100 m event at the Connecticut state track meet alongside her teammate, Terry Miller, who brought home a silver medal in that race. By all accounts, their victories were successes and the girls were poised to become promising young athletes who might even compete at the college level.

Yearwood and Miller's celebrations were cut short, however, when a parent of one of their teammates began circulating a petition during subsequent track meets calling for the Connecticut Interscholastic Athletic Conference to ban transgender athletes from competing. The cause was quickly picked up by far-Right pundits, such as Fox News' Tucker Carlson, and conservative interest groups, such as the evangelical Christian and anti-LGBT organization, Alliance Defending Freedom (ADF), which promoted the narrative that Yearwood and Miller represented an existential threat to girls and women's sports (Barnes 2020). These claims hinged on what commentators argued was an unfair biological advantage due to Yearwood and Miller being designated male at birth,[1] and in February 2020, three Connecticut families teamed up to file a federal lawsuit to prevent the girls from

---

[1] Throughout this paper: I use "designated/assigned at birth" to refer to the sex individuals are assigned at birth, which can differ from gender identity.

competing in the season's upcoming track meets (Barnes 2020). Although the COVID-19 pandemic cancelled all high school sporting events for the following year, compelling a federal judge to dismiss the case as moot, the controversy still received international attention and inaugurated a period of fierce and on-going debate regarding the proper place of transgender competitors in sports (Ennis 2021).

Yearwood and Miller are not the only young people caught up in battles of who, exactly, can compete on sex-segregated sports teams or which transgender people are eligible to receive care. With respect to athletics, dozens of bills seeking to formally ban transgender girls from athletic competitions were introduced across the states from 2020 through 2022, with 15 of them being signed into law (Movement Advancement Project 2022). These proposed laws range from complete bans on transgender athletes competing on the sports team that correspond with their gender identity, legislation allowing school districts to make decisions on transgender athletes, and requirements that transgender youth undergo medical transition—usually hormone interventions for one year—before they can compete[2]. Although the Department of Justice filed statements of interest on two such bans in Arkansas and West Virginia in June 2021, arguing that they violate the equal protection clause of the 14th Amendment, as well as Title IX, new bills prohibiting transgender youth from sports continue to be introduced or passed (Raby 2021).

These efforts to exclude transgender youth from sports echo other legislative efforts to make healthcare for young transgender people inaccessible and bar transgender people accessing public restrooms (Murib 2019). They also go hand-in-hand with mobilizations to criminalize protest activity, eliminate access to voting, terminate abortion access, and authorize the state to control how populations access medical care more generally. At the core of these mobilizations is the continuation of a decades-long effort by evangelical Christians and far-Right political actors to ensconce White, heteronormative, and (re)productive families as prototypical citizens and the bedrock of the nation. Having lost the battle over marriage equality, conservative political actors have turned their attention to transgender rights, seeking to criminalize and stigmatize transgender identification in the name of preserving the presumed complementarity of men and women (Minter 2017). In this view, the long history of projecting these heteronormative family configurations as the ideal citizens and the on-going ascension of far-Right discourse in the US means that we might expect even more versions of legislation seeking to divest transgender people of rights and opportunities to occupy public space in the future.

This paper examines how people understand these efforts to foreclose opportunities for transgender people to exist in public spaces, particularly with respect to transgender youth participating in school athletics. By focusing on discourse—how transgender identity and people are characterized and understood—it joins a growing body of research that interprets the text of legislation seeking to limit the rights of transgender people (Colvin 2007; Murib 2019; Sharrow 2021). That scholarship examines laws targeting transgender people and argues that the ultimate goal of these bills is to reify perceptions of binary sex as natural to maintain what Elizabeth Sharrow calls "cisgender supremacy", or the elevation of biology as determinant of gender, which remains static over time (Sharrow 2021).

This paper adds to this scholarship by investigating a relatively less understood factor shaping experiences for transgender people: how non-transgender people in the general public understand transgender identity and, with it, gender. I turn to YouTube for a number of reasons—namely that it is a major source of news for a significant number of Americans as well as one of the most popular social media sites—to answer these questions (Pew Research Center 2020). Using a grounded, iterative approach to discourse

---

[2]	Six states (TX, AL, NC, KY, ID, FL) require trans athletes to compete based on sex assigned at birth. Six states allow school districts to decide: AK, CT, GA, KS, PA, WI. Maine allows students to choose where they compete. NJ and NM requires evidence to prove transitioning (presumably medically transitioning). MI and OH have similar requirements based on time (1 year). OR allows trans boys to compete as long as they agree to be excluded from girls' teams. Trans girls who want to compete with other girls are required to undergo one year of hormone interventions before being allowed to compete.

analysis in combination with Automated Text Analysis, I analyze over 60,000 comments on three YouTube videos discussing transgender athletes in school sports to develop and test theories of how people understand transgender people in popular discourse. Three dominant themes emerge in this study of popular discourse on transgender athletes: a firm attachment to biology as determinative of both sex/gender and, by extension, athletic abilities; racialized understandings of who constitutes a proper "girl"; and perceptions of sex-segregated sports as the sole way to ensure fairness in athletic opportunities.

## 2. Background and Theoretical Frame

### 2.1. Transgender Politics

Political science scholarship generally finds that while there are a handful of predictors that determine support for transgender political appeals in the US, such as age and partisanship, there is also a diffuse lack of willingness to back transgender rights and transgender people. Partisanship is the strongest factor that shapes both voting for transgender candidates as well as the likelihood of endorsing pushes for transgender rights, with Democrats and liberals more likely to back transgender candidates and issues than Republicans and conservatives (Jones et al. 2018; Haider-Markel et al. 2017). These diverging views are driven to some degree by elite discourse, which has been shown to inform and shape attitudes about minority rights (McGraw et al. 1995; Pérez 2015; Schneider and Jacoby 2005). Studies highlight partisan differences in support for transgender rights and people speculate that these findings are a result of clearer messaging by Democratic candidates on transgender issues relative to their Republican counterparts (Flores 2015; Lewis et al. 2017). What emerges in these studies is focus on the role that education from a variety of sources, as well as awareness, plays in shaping widely held beliefs about transgender people that, in turn, influences support for a constellation of issues that impact the lives of transgender people (Flores et al. 2018). If, as this scholarship posits, the general public is largely unaware of transgender issues and has relatively simplistic views of gender as a static binary, then providing information about gender norms as social constructs might go some way in enhancing acceptance for transgender people.

The recent uptick in anti-trans legislation suggests that the inverse is also true: lack of knowledge about gender as a capacious, dynamic, and racialized social construct might also be exploited for political ends. When efforts to halt to same-sex marriage fizzed in the wake of the Supreme Court's 2015 decision in Obergefell v. Hodges, opponents mounted campaigns to block the extension of rights to transgender people by exploiting the presumed naturalness of sex segregation, and, by extension, binary understandings of gender. From 2014 through 2018, these mobilizations took the form of legislative bans on transgender people accessing public restrooms, which scholars argue are logical outgrowths of conservative mobilizations to preserve the constitutive parts of the heterosexual family: one male and one female, united for (re)productive ends and in the interest of preserving White racial purity (Minter 2017). Feminist thinkers underscore that it not just any heterosexual family unit promoted through the policing of sex fostered by anti-same-sex marriage legislation, but rather the on-going production of prototypical citizens, who in the US are White, reproductive, gender normative, and future workers (Alexander 1994; Collins 2001).

Taken in this historical context, anti-trans campaigns are less about targeting transgender individuals and more about fostering discourse that preserves ideas about gender as static and rooted in biological complementarity, as well as who constitutes a proper citizen to whom rights and protections are extended. Sociologists Schilt and Westbrook's (2015) analysis of campaign materials produced to advance these bathroom bans, for example, finds that proponents of these bans discursively reframed the issue away from legislating discrimination against transgender people and instead focused on reiterating and retrenching widely held beliefs about gender as a natural and self-evident binary to

justify why bathrooms are sex-segregated.[3] These include the assumption that (White) women are uniquely vulnerable to violence in public space and must be protected from certain (Black and Latinx) men, who are biologically predisposed to commit sexual assault due to racialized assumptions about innate drive. That transgender women are not men is beside the point for these campaigns.[4] Schilt and Westbrook (2015) explain of these bans, "Under this logic, they often conflate 'sexual predators' (imagined to be deviant men) and transgender women (imagined to be always male). This exclusive focus on 'males' suggests that it is genitals—not gender identity and expression—that are driving what we term 'gender panics'—moments where people react to a challenge to the gender binary by frantically asserting its naturalness (27)." Gender, in this conceptualization, is not a social construct, but instead being in possession of certain genitals, or in the more updated form: specific chromosomal configurations. The centrality of biological traits in discourse around bathroom bans aims to reassert the perceived innateness of binary gender by drawing attention to what most people accept as fact: biological difference as the basis for sex segregation.

Although bans on transgender people accessing public restrooms largely failed, proponents of conservative beliefs about binary gender and a White supremacist heteropatriarchal order have shifted their focus to the variety of social, political, and economic sites where sex segregation is presumed to be ontologically prior to social influences, namely sports, to present these campaigns as logical and self-evident. This reorientation is to be expected as sports are one of the most visible sites of sex segregation, with millions of spectators regularly cheering on men's and women's teams at all levels of competition. These routinized practices of sex segregation almost always result in what political theorist Heath Fogg Davis terms "sex identity discrimination", or harms stemming from decisions about who does and does not belong in the sex categories of male and female and, by extension, spaces organized around accommodating individuals who reflect what are presumed to be typical male and female characteristics. Since gender norms are explicitly racialized, granting White men and women a monopoly on what it means to be a proper man and proper woman (Snorton 2017; Spillers 1987), sex identity discrimination operates as "intersectional sex affinity, or where and with whom individuals belong in the racialized social scheme of sex (Davis 2017, p. 27)." As in the case of bathroom bans, protecting certain girls (i.e., White girls) emerges as a theme uniting mobilizations to ban transgender girls from athletic competition due to perceptions about fairness and safety that cannot be severed from deeply held cultural beliefs about the relationship between testosterone, aggression, and masculinity as well as the presumed inferiority of girls and women.

## 2.2. Gender, Race, and Hormones

Nowhere is the adherence to strictly biological understandings of sex/gender more evident than in the ways that people perceive the influence of testosterone on athletic ability and, by extension, masculinity (Fine 2017). Katrina Karkazis and Rebecca Jordan-Young refer to how people understand testosterone as "T talk", or the variety of ways that testosterone circulates as both a chemical substance and a multifaceted cultural symbol to prop up dominant White supremacist and heteropatriarchal ideologies that rely on the elevation of (White) men as naturally more powerful and dominant than women, who are perceived to be nurturing and passive (Jordan-Young and Karkazis 2019).

These lessons about testosterone and its effects start at a very young age and are often constructed at the site of youth sports and athletics. Studies of youth sports show how biological essentialism, or "T talk", shapes discussions about boys' and girls' athletic abilities and underscores that there is a recursive relationship between perceptions of boys' "natural" behavior and the construction attached to masculinity vis-a-vis testosterone.

---

3   The introduction of sex-segregated bathrooms in Western Europe and North America tracks alongside the entrance of women in the public sphere during the 19th century (Cavanaugh 2010).

4   Research finds that it is often transgender and gender nonconforming people who report being disproportionately harassed in public restrooms (Grant et al. 2012).

This logic follows a well-worn path: boys excel at sports because they are innately more competitive, aggressive, and less likely to cry, and testosterone is the reason for these differences when compared to girls, who are more sensitive to criticism and less invested in winning competitions, presumably due to a relative lack of testosterone (Karkazis and Jordan-Young 2018, p. 8). What sociologist Michael Messner's study of boys' little league teams reveals, however, is that these behaviors are scripted onto boys at a very young age by coaches and parents. On Little League teams, boys do cry when they are disappointed or frustrated, but are encouraged to do so in private. By the time they reach middle school, however, the boys have learned how to express their emotions differently: they throw gloves, kick the walls of the dugout, or, in some extreme cases that are tacitly condoned, talk back to the umpires to advocate for themselves (Messner 2011). These changes are the result of social influences, such as coaches expecting such behavior, even while well-meaning people will point to them as unquestionable proof of the material fact of gender distinctions between boys and girls (Enke 2012).

It is important to note that these displays of brute aggression, credited to testosterone, are only acceptable when White boys do them. Ann Ferguson's (2000) study of how Black boys are disciplined in public schools explains that, "though the racial etiquette of today's form of racism has sent a discourse of racial difference underground, it piggybacks on our beliefs about sex difference . . . . Black boys are constituted as different from boys in general by virtue of sexing racial meaning (23)." In other words, it is impossible to tease race apart from the perceived innocence of White boys' healthy and normal aggression and the construction of Black boyhood as characterized by unchecked and troubled aggression. Because meanings attached to gender are shaped against the backdrop of racial difference, White and Black girls are not excluded from these formulations. Robin Bernstein describes this process as one of constructing "racial innocence", in which childhood is used to signal the natural and therefore justified outcomes of racial projects in the US after slavery. "What childhood innocence helped Americans to assert by forgetting, to think about by performing obliviousness, was not only Whiteness but also racial difference constructed against Whiteness. Racial binarism—understanding race in terms of White and nonwhite, or a "Black and White" polarization that erases nonblack people of color – gained legibility through nineteenth century childhood (Bernstein 2011, p. 7)." Bernstein's analysis elaborates how the innocence of White children, particularly girls, was achieved through perpetuating the libel that Black children, especially boys, were immune to pain, to justify enslavement. These complex and on-going processes of racialization sutured to gendering consequently delimit what it means to be White and Black children, in general, but especially in the realm of athletics where bodies are a focal point of competition and sex segregation is taken as self-evident, natural, and logical.

*2.3. Fairness and Testosterone*

The meanings attached to certain gendered bodies in sport is made even more relevant by the proliferation of legislation across the states that seeks to ban transgender athletes—always transgender girls—from competing in girls' (and women's) athletics to preserve fairness (Sharrow 2021). Proponents justify these bans by drawing on popular narratives that celebrate Title IX of the Education Amendments of 1972 for levelling the playing field for girls and women in sports. Much like the discourse around proposed bathroom bans, which promoted the fallacy that those legislative efforts would preserve long-accepted ways of protecting women from men in public space, those in favor of banning transgender girls from sports point to Title IX to argue that sex segregation is a fireproof way to ensure fairness in girl's and women's sports.

These myths about Title IX's role in paving the way for girls and women to enter athletics and secure fairness in competition persist despite there being no mandate to create sex segregated sports in the text of the legislation. Passed in 1972 and at the height of battles over the Equal Rights Amendment, Title IX was motivated by an interest in preventing sex discrimination and differential treatment in higher education (Office of Civil Rights 1979).

Its provisions include items such as prohibitions against using sex as a way of distributing resources and opportunities—i.e., admission to colleges and universities that receive federal funding—as well as protections against harassment understood to be shaped by one's sex.

The evolution of Title IX as synonymous with women's sports evolved in response to opponents of the legislation who sought to stall it by stoking fears that its passage would result in the end of spaces organized by sex according to the social norms of 1972, including locker rooms and the military. Proponents of Title IX secured its passage by rebutting these anxieties—specifically with respect to sports—by arguing that implementation of the bill would instead seek to ensure parity amongst the sexes, not integration. This compromise in its simplest form means that colleges and universities are required to allocate resources in a comparable manner across men's and women's teams. As a result of on-going litigation since Title IX was passed in 1972, colleges and universities can also demonstrate adherence to the law by pointing a history of expansion in sporting opportunities for the underrepresented sex—usually women (Pickett et al. 2012). The current strategy for compliance entails expanding into sports such as women's lacrosse and ice hockey, which entail relatively higher expenditures and consequently go some way in demonstrating parity in resource allocation. Since these sports are more expensive, requiring significant investments in equipment, training, and space, favoring this tactic of compliance with Title IX means that it is disproportionately the White and wealthy women who train in these sports that benefit from the policy. Thus, while Title IX has succeeded in adding more women's sports teams and encouraging girls and women to train as athletes over its fifty year history, the single-axis approach to defining discrimination, "on the basis of sex", means that Black, Latinx, Asian, and Native athletes are closed out of sporting opportunities and scholarships because they tend to compete in relatively less expensive sports, such as track and field or basketball (Crenshaw 1989; Halley 2000; Pickett et al. 2012).

Furthermore, the long-term effects of Title IX's implementation and the creation of sex-segregated sports teams means that the "female athlete" is defined against boys and men, who are understood simply as "athletes" (Sharrow 2017). The perception that men are superior athletes is shaped and maintained by sporting regulations that presume girls and women are slower, weaker, and have less stamina, such as tennis regulations that limit women's sets to three (to men's five). Creating different rules for women's sports consequently requires separate divisions and forecloses opportunities for contests between women and men that could potentially disrupt the pretense that men are always better athletes than women, even while women compete at similar—or better—levels against men in a variety of sports that include distance swimming, golf, rock climbing, dogsled racing, fencing, skiing, snowboarding, shooting, and archery (Leong 2017). These trends underscore that although testosterone can correlate with speed and strength, it does not determine athletic ability in any absolute sense. And yet, the effect of testosterone is taken as a foregone conclusion, such that there is, to date, no research on the actual effects of the hormone on athletics—just anecdotal accounts of men's dominance (Jones et al. 2017; Leong 2017). Organizational decisions grounded in perceptions about these sexed differences have evolved as synonymous with Title IX, which, in turn, shapes how fairness is understood as efforts that ensure opportunities to win. This focus on winning ignores the wide variety of reasons that people compete in sports such as camaraderie, physical and mental wellness, developing skills, the thrill of overcoming challenges and meeting goals, and fun (Simon et al. 2015). It also elides the ways that class and access to resources skew fairness in favor of those who can afford the time, equipment, and coaching necessary to win competitions.[5] Elevating winning over these aspects of competition and training shapes how the public understands sports, fairness, and who is allowed to compete, with sex segregation doing most of the heavy lifting when it comes to ensuring fairness. This

---

[5] To this point: there are almost always variations in class across public school districts in the US, and yet teams from high schools regularly compete against each other without explicit commentary about how relatively more wealthy schools fare better than schools serving economically precarious student populations.

attachment to sex segregation as fairness has reached a fevered pitch in the public sphere, as athletes born with intersex conditions, such as Caster Semenya and Dutee Chad, are barred from competition and transgender athletes enter athletic competitions faced with stigmatizing rhetoric based on the perception that these athletes imperil fairness.[6]

To investigate the ways that these constructions of who, exactly, constitutes a proper girl and, by extension, a beneficiary of Title IX and thus fairness, I turn to popular discussions that take place on social media, specifically YouTube. Studying the conversations that take place in conjunction with YouTube videos reporting on transgender athletes and legislative efforts to ban them from sports serves as a window into how people with varying backgrounds and levels of knowledge understand transgender people. Do these viewers support the civil rights of transgender people while simultaneously rejecting the presence of their bodies in the public sphere as the literature suggests? How are transgender athletes, specifically transgender girls targeted by the bans, understood by viewers of videos about transgender athletics? In what ways do prevailing myths about the effects of testosterone structure the discourse on transgender athletes, limiting or opening opportunities for them? Finally, in what ways are understandings of girls' athletics shaped by raced and gendered constructions of what it means to be a proper girl? The following section outlines my analytic approach to answering these questions.

### 3. Data and Method

There are six reasons I focus on YouTube comments in this study: (1) They capture perspectives on issues held amongst the general video viewing public because anybody can comment on a video. Although YouTube is available to anybody with an internet connection and the attention span to consume videos, it should be noted that YouTube data are not a substitute for other forms of data, such as surveys. That said, the findings generated can only be generalized to the YouTube viewing population. (2) Data on viewership are clear, allowing researchers to state the relative importance and influence of a video. In this study, I sample videos that have at least several thousand views, which I interpret as an indicator of the importance as well as the reach of the video in shaping and reflecting the views of YouTube users. (3) YouTube comments provide real-time insight into how people perceive social and political problems, how they debate them, and what vocabularies and framings they invoke to advance their views. (4) The "like" and "dislike" functions give insight into the extent to which certain viewpoints are embraced by other viewers. (5) Opportunities to view the correspondence between the content of the video and the views taking shape (i.e., potential priming effects, broadly construed). (6) A growing number of adults in the US report receiving their news primarily from YouTube (Pew Research Center 2020). While there is limited information available on who, exactly, receives their news from YouTube, research shows that user behavior in the comments on YouTube videos demonstrates a strong correlation with user behavior on other social media sites such as Facebook (Bessi et al. 2016). YouTube data therefore provide a useful snapshot of social media consumption and engagement.

In this study, I sample three videos that discuss transgender athletes competing on girls' teams and emerging legislative efforts to ban them from sports that began in 2018 with Yearwood and Miller's winning streak on the girls track team in Connecticut. I identified these videos by drawing on my knowledge of transgender politics as it pertains to athletics as well as interest groups that have been active in opposing or defending transgender athletes in youth sports. After developing search terms to identify neutral, negative, and positive discussions of transgender athletes, I sorted the results by number of views, which

---

6　Caster Semenya and Dutee Chad are track athletes who have been subjected to sex testing over the duration of their careers as track athletes. That both athletes represent countries in the Global South in international competition and have been barred—and stigmatized—from competition is pointed out by some scholars as indicative of the extent to which who constitutes a proper woman is a racialized concept, with White women always serving as the measure against which all other women are compared (Karkazis and Jordan-Young 2018).

I use as a proxy for the influence and scope of each video. I do not have demographic data on viewers (location, age, race, sexuality, gender identity, or partisan affiliation), and future scholarship using YouTube data would benefit from developing ways to extract this information. The videos, search terms, and number of views are summarized in Table 1.

**Table 1.** YouTube Video Titles, Search Terms, and Viewing Data.

| Video Title | Search Term | Date | Views | Comments |
|---|---|---|---|---|
| ABC News | "Andraya Yearwood" | 18 June 2018 | 2.5 million | 49,960 |
| ADF | "Alliance Defending Freedom transgender" | 8 September 2020 | 1 million | 6698 |
| ACLU | "ACLU transgender athletes" | 15 May 2021 | 138,667 | 2443 |

Source: YouTube.

The first video I examine is one that I consider a neutral report on the issue as it attempts to present many different sides and does not advocate a particular policy response; it rather presents the facts of efforts to ban Yearwood and Miller from competing and features interviews with them alongside interviews with the parents petitioning to ban them from sports. This video is classified as neutral by virtue of presenting "both sides" of the issue, in the style of conventional journalism for broad audiences with the goal of education. On 18 June 2018, ABC's Good Morning America aired a segment on Yearwood and Miller that, to date, has received 2.5 million views on YouTube and 49,960 comments.

The ABC segment was followed by several other videos on the same topic, including one published by the conservative think tank, Alliance Defending Freedom (ADF), on 8 September 2020 titled "Should 'Transgender Women' Be Allowed to Compete in Women's Sports?" with just over one million views and 6698 comments. I categorize this video as a negative portrayal of transgender athletes as ADF has been one of the main organizations mobilizing opposition to transgender athletes. This is reflected in the content of the video, which features commentary aimed at denying transgender identification and stigmatizing rhetoric to justify opposing transgender athletes in youth competitions.

Finally, I analyze comments from a video created by what I consider to be an ideologically left-leaning interest group, the American Civil Liberties Union. I classify this video as positive, as the ACLU has taken on an active and prominent role in defending against efforts to ban transgender athletes from sports. The video in this study reflects this orientation, featuring a dad who describes his struggle to accept his transgender daughter and his commitment to protecting her. That video, titled "Missouri Dad Testifies Against Trans Youth Athlete Ban", was uploaded on 15 May 2021 and has been viewed 138,667 times with 2443 comments.

I draw on two empirical strategies for examining this unique dataset of close to 60,000 YouTube comments across three videos: the first is a grounded, interpretive reading on a sample of comments that includes the top 500 "liked" comments and 100 comments randomly selected to assess the degree to which the top 500 comments reflect sentiments among users. The second is Automated Text Analysis of the entire corpus, with each video comprising the unit of analysis.

Since the amount of data in this study would take a significant amount of time for a team of coders to process, as a first step, in this paper, I sort comments by "likes" and adapt Glaser and Strauss' 1967 Grounded Theory Method (GTM) to conduct iterative coding of the top 500 comments for each video[7] as well as a random sample of 100 comments outside of the top "liked" comments. GTM differs from typical positivist or quantitative approaches in that the researcher does not begin analysis with hypothesis development and testing, but rather proceeds inductively to generate data-driven theories and explanations of phenomena being studied. Conducting iterated coding of the comments, with an eye for themes that emerge in first, second, and third readings of the comments helps to generate

---

[7] The "top 500" are determined by sorting the comments by "likes" and taking the first 500.

theories about how people discussing transgender youth in sports understand the issue as key themes begin to repeat across the texts examined—in this case, comments. Since research shows that there is a limited range of information about transgender people and issues amongst the general public, I expect these comments to cluster into related themes that capture how people discuss and understand transgender people (Flores et al. 2018; Lewis et al. 2017). To address the possibility that the top "liked" comments are not representative of genuine views but are instead framed polemically to attract "likes", a research assistant and I coded a random sample of 100 additional comments that did not receive a significant number of "likes".

I then use Automatic Text Analysis (ATA) to investigate latent connections amongst terms found in the YouTube comments for each video and compare them against the themes that are revealed in the GTM.[8] This second step allows me to confirm (or reassess) findings from the GTM due to ATA's and Natural Language Processing power as a facet of grounded theory, in which the researcher uses machine learning to conduct semi- or fully automated analysis of text data with the benefit of achieving near-human precision in a fraction of time (Banks et al. 2018). One way to think about the power of ATA is as a facet of factor analysis, which identifies latent connections amongst terms (Grimmer et al. 2021). I expect that the comments, taken in the aggregate, will largely confirm the themes found in the hand-coding. I also anticipate that it will reveal additional avenues for investigation given the breadth of that analysis.

I report the findings of these analyses in the following three sections: the first describes my findings from the GTM analysis by dominant themes and discusses them in relation to the scholarship on gender and sexuality in US politics. The next section briefly reports my findings from the ATA analysis, followed by a discussion. I then conclude by reflecting on what this analysis illustrates and offer suggestions for future research.

## 4. Findings: Grounded Approach

Analyzing the top 500 comments that received the greatest number of "likes" for each video reveals three prevalent themes: (1) a firm attachment to biology as determinative of both sex/gender and, by extension, athletic ability, (2) racialized assertions about who is considered a proper girl, and (3) sex segregation as the only way to ensure fairness in youth athletics. These themes emerge regardless of video type—neutral, positive, or negative—suggesting that there are relatively consistent ways that the transgender people and athletes are understood amongst the YouTube users who viewed these videos. For this reason, I present the findings for this stage of the analysis by theme, not video. Current research on YouTube users confirms this intuition about comments as a type of echo chamber, finding that although some users might change how they engage with content to take on an alternate viewpoint at one point in time, all subsequent comments will be aligned with the new position to produce another echo chamber. Although I present the themes separately, each of these themes is shot through with logic that questions the authenticity of transgender identification as well as racialized understandings of who constitutes a proper girl. Taken together, these comments paint an overall bleak picture for how the public understands transgender people and girls' (and, by association, women's) athletic abilities. I take each of these themes in turn below.

### 4.1. Biology: Athletic Ability and Destiny

The most common way that people commenting on the YouTube videos understand transgender youth participating in sports is through the lens of biology. These comments converge on two dominant themes: biology as necessarily corresponding with athletic ability, and biology as static, which is illustrated through analogies to race as well as inanimate objects. Across these themes, biology functions as a screen to obscure the role of

---

8    ATA can be best understood as a facet of factor analysis in which latent connections across words are identified (Banks et al. 2018; Grimmer and Stewart 2013; Grimmer et al. 2021).

social factors that do a variety of things, including: inform gender norms, influence how sports are organized, and give shape to athletic ability. This allows commenters to pose their opinions on transgender athletes as biological and neutral scientific facts as opposed to perspectives informed by White supremacist and patriarchal notions about the proper roles of men and women.

For example, in many comments, biology is posited as overshadowing all other influences due to the presumably static nature of unseen and innate factors such as chromosomes and hormones, or "T talk". One of the most "liked" comments in the ABC video explicitly posed biology against social factors, stating, "Biology over sociology. The average biologically born male will always be faster and stronger than a biologically born female. Point, blank, period." Another commentator followed in this vein to enumerate the various ways that biology shapes differences between men and women by cataloguing anecdotal evidence of variations in bodies:

> No advantage? NO ADVANTAGE!? Let me drop some knowledge on you.
>
> 1. Men have bigger hearts than women. (able to pump more blood)
>
> 2. Men have bigger lungs than women. (able to produce more oxygen at maximum exertion levels)
>
> 3. Men have a greater amount of muscle bulk than women. (more power)
>
> 4. Men's legs are about 80% muscle, women's legs are about 60% muscle. (again, more power)
>
> 5. Men produce far more testosterone than women. (advantage is many ways)
>
> 6. Men have more and larger fast twitch muscle than women. (helps with sprinting)
>
> 7. Men have more hemoglobin than women do (protein in red blood cells that carries oxygen to the body's tissues, including muscles)
>
> I could name many more, need I !? It is not fair for women to be competing against transsexuals. PERIOD.

Here, the individual making the comment links what is taken to be natural—biological differences in the size of bodies, muscle mass, and hormones that are presumed to map onto sex assigned at birth—and links them to athletic ability, ignoring that not all athletic competitions require size, speed, or lung capacity. Even while it is likely that most of the people reading these comments are not elite athletes (or even recreational athletes) and therefore know quite well that athletic ability requires more than sheer size or mass, the language of comments such as these, which "drop some knowledge", poses readers as recipients of logical conclusions about the correspondence between larger hearts and lungs, or certain types of muscle mass, and performance in sports. It is rarely the case that one hears athletes honored exclusively for their superior size in any sporting competition, yet these claims achieve their vehemence through the implicit and explicit denial of factors that are empirically demonstrated to shape athletic ability. These include access to resources (including nutrition for building muscle mass), technological innovations in sporting equipment that enhance outcomes, training, and—perhaps most surprisingly—talent, which is often celebrated in stories of athletic triumph. The picture of athletic ability that emerges in these comments is one that is attached to bodies with testosterone.

Another way that commenters referring to biology attempt to seal their arguments off to critique is by drawing analogies to other forms of presumably biologically informed differences, namely in skin tone and, by association, race. In other instances, biology is posed as static and irrefutable through comparisons to biological variations between species. Across these types of comments, readers are meant to understand the fixed and determinative nature of sex assigned at birth as a facet of biology, which means that the authors of these statements are not simply weighing in on the question of transgender girls participating in sports; they are making epistemic arguments about the very existence of transgender identity and experience.

With respect to race, the logic follows a familiar and yet simplistic pattern in the comments: since Black people cannot change the color of their skin tone, which is posed as rooted in biological factors such as the presence or absence of melanin, men and women are also unable to alter their gender identities due to the constraints of biology, such as chromosomes or genitals. Statements such as these are often posed as absurd to underscore the self-evident nature of the criticism directed at transgender athletes. For example, one author speculates, "If Mike Tyson says he identifies as a woman and steps in the ring as a woman against a biological woman, what would be said?" Mike Tyson appeared in several comments, with another writer satirically stating, "Breaking news: Mike Tyson came out as a female. Now he is competing on a female boxing tournament. Me: wow what a massacre." In both comments, the authors draw on shared knowledge of Tyson as a boxer with a history of battering women to equate Yearwood and Miller with elite athletes as well as men with abusive histories. Comments such as these also evoke associations with Black boyhood as troubled and a threat to White innocence, especially as it is understood as attached to White girls.

These themes are also evident in the 100 random comments coded outside of the most liked statements across the videos. For instance, in the ACLU video, one author stated, "Gtfo [get the fuck out] with that bullshit! I don't give a fuck what people say, men should not compete in women's sports. PERIOD. Just imagine Lebron [sic] James in a wig joining the WNBA because he wants to classify himself as a women [sic]." Although none of the commenters explicitly identify race as the logic informing claims such as this, the frequency with which figures such as Mike Tyson, Kevin Durant, LeBron James, and Shaquille O'Neal (a 7′ 1′′ championship basketball player) are cited to make these points— and to the exclusion of White athletes—implicitly stitches racial difference to athletic ability, and maps those differences onto gender, which is taken as glaringly obvious due to its presumed grounding in biology. Across these comments, purportedly biological logic like this helps to shield commenters from charges of racism and sexism.

In other instances, comments were much more explicit about the racial dynamics at play. The defensiveness in these statements indicates the extent to which it is impossible to tease apart the race of the transgender athletes featured in the ABC news story, gender, and the opposition fomented in the discussions posted below videos. For example, one author responding to the ABC segment argues:

> Parents say transgender black girls should be forced to compete as boys. I love how they just threw "black" in there. Guaran-damn-tee [sic] none of the parents were complaining about that aspect, they just decided smear them a little more to make them look racist as well as transphobic when they just want what's best for their daughters.

Here, the author evacuates Yearwood and Miller of their racial identities to defend the parents opposing their participation from allegations of racism. According to the author of this comment, the parents opposing Yearwood and Miller are neither transphobic, nor racist, but rather good parents who simply want to see their daughters succeed. These parents are also all implied to be White by citing allegations of racism as the reason for the defensive response. Within the logic of comments such as these, to support those racially unmarked daughters is to defend against the threat posed by those who are constructed as unnatural and outside of childhood: transgender (Black) girls. Taken together, readers are meant to understand from the comments about Mike Tyson and racism that girls need protection from their parents against non-transgender boys, which also locates girls (and women) as innately inferior to boys and men in athletic competition. This logic reifies patriarchal views on the subordination of women and takes their relative lack of skill, strength, and athletic acumen as natural facts of gender. It also constructs transgender youth as outside the realm of parental care, as if identifying as transgender disqualifies young people from the care of concerned parents.

The theme of parents protecting vulnerable daughters also plays out in comments that viewed the obstacles facing transgender athletes sympathetically. For instance, the ACLU

video features the father of a transgender daughter evocatively speaking about his struggle to accept her identity and asserting his commitment to protecting her right to compete based on his love of her. A significant number of the most "liked" comments celebrated this father's commitment to his daughter and drew on similar logics of protecting vulnerable girls. One commenter stated:

> I wish all transgender children had fathers like this. A father that loves them unconditionally. A father that doesn't let society decide whether his child should be happy or not. A father that is willing to stand up against bullies. A father who is willing to proudly say that yes, this is my child. They are transgender. And still love, support, and accept them. A father who won't make empty promises. One who will push aside his own insecurities and fears, to allow his child to navigate the world happier. One who will protect his child while the child does so. I wish my father was like this. But he's not. So here I am, crying, watching a video of a father I'll never have.

Comments such as these flip the script on parental care, applying it to transgender children, who, according to this author, are often most in danger of parental rejection. In so doing, it also preserves the perception that young girls are vulnerable and in need of protection by stronger, older men—in this case, a caring father.

In another running theme, commenters made frequent use of analogies to animals, inanimate objects, and mythological creatures to make the case against transgender athletes and the legitimacy of transgender identity. At first blush, many of these comparisons strike a decidedly bizarre tone. These comparisons appear in comments made on all of the videos, with one representative comment stating, "I'm going to enter my pet cheetah into the Olympics. It identifies as human." Another asks, "Why don't the real girls identify themselves as motor bikes and race the transgender with motorbikes." A third advises the following, "The women should identify as centaurs and ride horses on the track." Although it would be easy (and quite rational) to dismiss comments such as these as the type of vacuous nonsense that fills comment threads on websites, statements such as these take on an especially sinister air when viewed alongside the robust discussion about opportunities for transgender girls to participate in sports. Put differently, comparisons such the ones rank in the top 500 "most liked" comments in a discussion of transgender athletes that also poses "scientific knowledge" as justifications for discrimination, regardless of whether or not these comments are "liked" for their inflammatory nature. Drawing analogies between transgender girls and cheetahs, motorbikes, or a strange amalgam of centaurs that ride horses consequently accomplishes two simultaneous moves: epistemically erasing transgender girls from consideration by posing them as artificial and logically impossible, with the collateral effect of dehumanizing them. What reads like humorous speculation about transgender girls justifies foreclosing protection and opportunities for transgender youth entirely.

*4.2. Fairness and Sex Segregation*

Another dominant strain in the comments focuses on reinforcing sex-segregated sports as the only way to ensure fairness for non-transgender girls and women. In many cases, these comments cement the imagined links between biology, gender, and athletic ability by posing transgender girls as athletic interlopers who seek to exploit presumed biological advantage to dominate girls' sports. This circular logic is evident in several of the comments through comparisons to other bodily differences that are taken to be natural and uncontested. In one example, an author makes a comparison to people with disabilities, explaining that: "This is like an able-bodied person claiming they 'feel' disabled and being allowed to compete in the Paralympics. Ridiculous." Another writer stated, "I identify as a disabled person now if you'll excuse me let me go and dominate the Special Olympics." These comments, and others like them, pose gender as a static and "true" aspects of bodies, which in turn collapses transgender identity as false and associated with feeling states that can be adopted at will to deceive. The perceived artifice of transgender identification is

achieved through the analogy to disability understood primarily as physical differences. Since readers presumably agree that specialized, segregated competitions are one way to ensure fairness for athletes with disabilities, the logic follows that transgender girls competing against non-transgender girls denies them of opportunities to win, in much the same way that competition between people who are differently abled might be unfair to people with disability. These analogies are used to argue that any sporting competition that allows transgender girls to participate is necessarily unfair to non-transgender girls.[9] Locking transgender girls into their sex assigned at birth—as boys, or males—poses "real girls" at a perpetual disadvantage to boys and justifies sex-segregated sports based on the logic that girls can never compete against boys and win. Fairness, in this view, is only achieved by segregating sports based on biological factors, such as gender, which is accepted as purportedly self-evident as physical disabilities.

Segregating sports by gender is cemented as a solution in the comments, with several people suggesting that the best way to address the question of transgender athletes would be to create a separate transgender league to ensure fairness for all involved. These comments generally cluster around the proposal that gender identity for girls and boys is static, while transgender identity is a third, exceptional type of identification that merits its own level of competition. One comment opined at length:

> It's fine that they are Transgender, but it is really totally UNFAIR if they're going to compete with natural born female athletes. I just think to make less controversial.. they should have sports for only transgender athletes or have different divisions for them to compete with each other. So male to female transgender should ONLY compete with another male to female transgender athlete and vice versa. I think that will resolve the problem and it's 100% fair.

Another commenter on the ADF video built on this theme and made use of analogies to make the point, speculating, "I mean if they really dont [sic] want the unfairness in this, they can just start a trans nba like how the start a national wheelchair basketball association." In other cases, the inferior status of non-transgender girls is implied to justify creating sports leagues, "Why don't they make a specific category for the trans women and trans men (both separate of course). This isn't fair, this is a blatant disregard for all the biological women who participate in sporting events." Fairness can only be achieved through separation in this statement, posing discrete divisions and competitions as self-evident solutions to the problems presented in the videos on transgender athletes.

Across these comments and others like them, the proposal for separate transgender athletic leagues echoes the reasoning behind the provision of "gender-neutral" bathrooms. Gender neutral restrooms create a third space outside of the conventional sex segregation of bathroom spaces that are most often found in shopping malls, airports, and university campuses and appear as restrooms designated for families and people with disabilities. In creating that third space, however, gender neutral bathrooms leave the division of public spaces by gender unquestioned. This effectively retrenches understandings of binary gender as normal and natural, and designates all those who do not conform to masculine and feminine gender norms as "other" (Enke 2012; Murib 2019). Proposing a separate sports league for transgender people accomplishes the same by creating a third space outside of "normal" leagues to preserve fairness for non-transgender girls while also scripting outsider status onto transgender youth. Studies of LGBTQ youth in sports find that transgender youth, in particular, struggle to participate in youth athletics, with the majority in one study reporting that they choose to not participate in sports due to harassment (Clark and Kosciw 2022).

Imagining sex-segregated sports as the only way to ensure fairness for non-transgender girls is most evident in comments about the question of transgender boys competing in

---

9　Although many embrace separate competitions for people with disabilities, there are also activists and athletes who call for integration. Examples include runner Oscar Pistorius of South Africa, who ran in the 2012 Summer Olympics.

sports. Once again, biological differences that are imagined as mapping onto sex assigned at birth are shored up in these arguments, with people assigned female at birth always occupying an inferior status relative to people assigned male at birth. One commenter speculates, "Why do you not hear about biological females transitioning to male beating male athletes? Because it never ever would happen." One writer allows for the possibility of transgender boys competing, but states that they will always do so at a disadvantage because, "You don't see transgender males shattering records in male's sports for a reason", with the reason being presumably the lack of innate testosterone. Another observes, "You notice how there are no male athletes complaining about trans men having an unfair advantage." These comments do not consider the possibility that transgender boys are able to compete at the same level as non-transgender boys by virtue of being assigned female at birth, even while transgender boys might also access hormone interventions, such as testosterone, which most of the comments recognize as conferring special athletic ability on those assigned male at birth (who might identify as transgender girls). These inconsistencies in logic point to the fallacy of biology and hormones as the reason for sex-segregated sports to ensure fairness for girls, revealing a project aimed at preserving gendered hierarchies in sports and society more broadly.

I now turn to the Automated Text Analysis to examine any correspondence between these themes and those identified by machine leaning in the larger corpus of text.

## 5. Findings: Automated Text Analysis

For this part of the study, I conducted a version of ATA in R that uses latent Dirichlet allocation (LDA) (Banks et al. 2018; Blei 2012). LDA is useful because it conducts machine learning based on the assumption that words will cluster into a certain number of topics present in the document (Banks et al. 2018). Each video constitutes the unit of analysis—or document—in this part of the study. I present ten topics revealed by the ATA and each of the terms that cluster within those topics in Table 1. I follow this presentation of findings with a discussion that puts these findings into conversation with the findings from the grounded analysis in Section 4.

ABC News

Table 2 reports the findings for the ATA analysis of 50,000 comments made on the ABC News video reporting on opposition to Yearwood and Miller competing on their high school track team. I ran three models with varying numbers of topic areas to assess which would provide the best "fit" for the data, arriving at 10.[10]

**Table 2.** ATA Results for ABC News Video Comments by Topic.

| Topic 1 | Topic 2 | Topic 3 | Topic 4 | Topic 5 | Topic 6 | Topic 7 | Topic 8 | Topic 9 | Topic 10 |
|---------|---------|---------|---------|---------|---------|---------|---------|---------|----------|
| "trans" | "people" | "male" | "now" | "like" | "testosterone" | "s" | "boys" | "like" | "woman" |
| "sports" | "like" | "female" | "right" | "lol" | "still" | "t" | "run" | "feel" | "man" |
| "compete" | "one" | "unfair" | "world" | "fucking" | "male" | "don" | "track" | "get" | "gender" |
| "transgender" | "trans" | "advantage" | "equal" | "shit" | "hormone" | "re" | "girl" | "want" | "can" |
| "athletes" | "think" | "females" | "rights" | "dudes" | "advantage" | "m" | "team" | "feelings" | "sex" |
| "fair" | "know" | "biological" | "equality" | "fuck" | "hormones" | "can" | "boy" | "parents" | "born" |
| "women's" | "said" | "males" | "god" | "two" | "even" | "trans" | "race" | "care" | "like" |
| "sport" | "say" | "compete" | "want" | "look" | "muscle" | "unfair" | "compete" | "dont" | "change" |
| "make" | "also" | "transgender" | "going" | "black" | "body" | "fair" | "school" | "hard" | "identify" |
| "competition" | "actually" | "born" | "get" | "stupid" | "therapy" | "think" | "can" | "happy" | "never" |

ADF

The results from the ATA of comments posted on the ADF video are reported in Table 3.

---

[10] I followed (Banks et al. 2018) recommendations for Text Analysis in R. This entails going through several steps to prepare the text, including eliminating symbols, making all words lower-case, removing all empty spaces, and all "stop words". Additionally, due to inherit limitations in social media data for text analysis, I developed a dictionary of terms, letters, and symbols to be removed from the corpus; all the same, the findings reveal that some of these "useless" words and letters remained in the dataset. I was unable to "stem" the words due to the uniqueness of social media data.

**Table 3.** ATA Results for ADF Video Comments by Topic.

| Topic 1 | Topic 2 | Topic 3 | Topic 4 | Topic 5 | Topic 6 | Topic 7 | Topic 8 | Topic 9 | Topic 10 |
|---------|---------|---------|---------|---------|---------|---------|---------|---------|----------|
| "?" | "woman" | "women" | "u" | "just" | "male" | "female" | "girls" | "unfair" | "people" |
| "s" | "man" | "men" | "people" | "get" | "testosterone" | "transgender" | "boys" | "advantage" | "gender" |
| "t" | "can" | "sports" | "now" | "people" | "advantage" | "compete" | "team" | "lol" | "trans" |
| "don" | "like" | "trans" | "right" | "feelings" | "still" | "male" | "run" | "like" | "just" |
| "re" | "just" | "compete" | "world" | "like" | "female" | "sports" | "track" | "fucking" | "can" |
| "m" | "girl" | "women's" | "black" | "feel" | "body" | "females" | "compete" | "dudes" | "sex" |
| "–" | "say" | "equal" | "god" | "hard" | "hormone" | "athletes" | "girl" | "shit" | "think" |
| "doesn" | "identify" | "rights" | "just" | "care" | "even" | "males" | "race" | "just" | "like" |
| "d" | "feel" | "equality" | "know" | "happy" | "hormones" | "biological" | "fair" | "fuck" | "one" |
| "won" | "born" | "competing" | "want" | "really" | "muscle" | "trans" | "two" | "stupidy" | "change" |

ACLU

The results of the ATA on the ACLU dataset are reported in Table 4.

**Table 4.** ATA Results for ACLU Video Comments by Topic.

| Topic 1 | Topic 2 | Topic 3 | Topic 4 | Topic 5 | Topic 6 | Topic 7 | Topic 8 | Topic 9 | Topic 10 |
|---------|---------|---------|---------|---------|---------|---------|---------|---------|----------|
| "testosterone" | "just" | "?" | "people" | "lol" | "people" | "female" | "men" | "girls" | "women" |
| "male" | "get" | "s" | "gender" | "u" | "like" | "male" | "women" | "boys" | "trans" |
| "still" | "now" | "t" | "just" | "like" | "want" | "unfair" | "woman" | "run" | "sports" |
| "advantage" | "school" | "don" | "can" | "fucking" | "just" | "advantage" | "man" | "compete" | "men" |
| "body" | "one" | "re" | "like" | "shit" | "feel" | "transgender" | "can" | "girl" | "compete" |
| "even" | "place" | "m" | "one" | "fuck" | "feelings" | "females" | "equal" | "team" | "women's" |
| "hormone" | "track" | "–" | "sex" | "black" | "can" | "biological" | "like" | "boy" | "fair" |
| "hormones" | "time" | "doesn" | "know" | "dudes" | "care" | "compete" | "now" | "race" | "transgender" |
| "female" | "going" | "d" | "said" | "world" | "get" | "males" | "equality" | "fair" | "people" |
| "muscle" | "first" | "won" | "think" | "stupid" | "right" | "born" | "just" | "guys" | ""competing" |

## 6. Discussion

The findings from the ATA offer insight into the overall trends in the comments. Perhaps most importantly, these findings largely correspond to the themes generated in the first step of the analysis in this study. I will take the results for each video in turn, beginning with the ABC (neutral) findings in Table 2. For instance, topic one indexes words that convey that sex segregation and the creation of a transgender-specific league might be the best way to ensure fairness. Topic 3 corresponds to how people discuss fairness in girls' sports generally: by posing transgender girls (and non-trans boys) as threats to girls' opportunities to succeed using words such as "unfair", "advantage", "biological", and "born". Similarly, topics 6 and 10 confirm the firm attachment to biology by revealing latent connections between words such as "testosterone", "muscle", "body", and "advantage" in topic 6 and "gender", "born", "change", and "never". Recall that one of the main ways that comments in the GTM analysis aimed to erode the legitimacy of transgender identification by posing it as a choice that is based on feelings, which are understood to be frivolous in comparison to the hard, scientific truth of biology. This theme is reflected in topic 9 and the association with "feelings" and "care". Finally, the ways in which these comments are shaped by racialized logics of gender and athletic difference—understood in explicitly racist ways—are revealed most starkly in topic 5, which identifies the latent connections between derisive terms, such as "lol", "shit", "dudes", "look", "black", and "stupid" in the text of the comments made on the ABC video.

The results of the ATA for the ADF video in Table 3 (negative portrayal of transgender athletes) confirm both the findings from the GTM as well as the findings from the ATA of the ABC News video. Here, again, linkages are identified across the terms "male", "testosterone", "advantage", "muscle", and "body". These terms correspond to themes identified in the GTM that emphasize the biological nature of sex and link hormones together with advantage to allude to issues of fairness. Topics 5 and 10 correspond to both the ATA and GTM for the videos in the aggregate, showing that people commenting on videos of transgender athletes convey a degree of skepticism regarding transgender identification. This appears in words such as "think", "can", "sex", "feelings", and "change". Finally, unlike the ATA results for the ABC and ACLU videos, the comments on the ADF video do

not link race, i.e., "black", to pejorative terms. This is a surprising finding that suggests that viewers of the ADF video were more likely to avoid discussions of race in conjunction with transgender athletes. While the data do not provide information about why race was avoided in this video, the defensiveness regarding race amongst opponents of transgender athletes from the GTM analysis could be drawn on here to suggest that these topics were avoided potentially out of concern for being accused of racism.

Similar trends emerge in Table 4, which reports the results of the ATA on the ACLU dataset. Recall that this is a video with a positive portrayal of a parent accepting his transgender daughter. Even so, topics 1 and 7 confirm that terms such as "testosterone", "advantage", "muscle", and "unfair" share underlying assumptions within each topic, thus linking them together. These terms reflect the findings of the GTM that shows that one of the dominant themes to emerge is the perception that testosterone shapes athletic advantage by shaping muscle development. It is noteworthy that terms generally accepted as conveying sex/biology, such as male and female, cluster in this topic. The appearance of "born" in topic 7 suggests that there are comments that convey sex/gender as innate and thus static. Conversely, the use of gendered terms such as "man" and "woman" hang together with both "equal" and "equality" in topic 8, indicating that these gendered terms were deployed when discussing separating teams out by gender, and the process of doing so as a way to ensure equality.

Taken together, these analyses reveal a relatively bleak picture for transgender athletes, and especially transgender girls, who wish to compete in sports. The findings show that understandings of biology as solely determinative of both sex/gender and athletic ability work hand-in-hand with perceptions of sex segregation as the only way to ensure fairness to foreclose opportunities for young transgender people in sports. Of particular concern is the ways that transgender girls are perceived in discourse on youth athletics. "T talk", or cultural myths about the effects of testosterone on bodies, imperils transgender girls in two ways.

The first is by locking them into their sex assigned at birth by virtue of unseen factors, such as chromosomes and hormones. These perceptions endure even while conventional approaches to gender confirmation in girls proceeds by first blocking testosterone production and subsequently introducing estrogen therapies. Underscoring these medical steps is not meant to further retrench the view that hormones correspond to athletic ability; rather, I highlight them here to illustrate that the claims about the enduring effects of testosterone made in the comments to these videos are misinformed opinions at best. They are rather "T talk", or assumptions based on the power of testosterone to shape athletic abilities. The oversimplified way that "science" is deployed in these comments suggests that there is a long way to go to confront the lack of understanding about transgender identity and embodiment and replace those myths with education about transgender identity.

Second, comments that infer a necessary link between testosterone and athletic ability harm both transgender and non-transgender girls by locking all girls into inferior status relative to those who are assigned male at birth and live their lives as boys and men. This logic asserts that sports are segregated by sex because boys (and men) are naturally stronger, faster, and more athletically able than girls (and women). Defending sex segregation on these terms confirms patriarchal ideas about the subordinate status of girls and women relative to boys and men. This shows up most obviously in the easy way that comments juxtapose "real girls" or "natural born girls" against the perceived artifice of transgender identity, which is derided as a "feeling" adopted at will. Preserving opportunities to win as the most important priority achieved via sex segregation cements this subordinate status into place, as evidenced by the frequency with which sexed terms—such as "male" and "female"—and "unfair" and "compete" appear together across each of the ATAs.

Finally, these statements against transgender athletes are grounded in racist appeals to biological difference that foreclose Black transgender girls (and boys) from gender. This is evident in the analysis presented in Section 4, which details how these comments confirm myths about presumably innate aggression of Black boys. It is also confirmed in the ATA

findings, in which words such as "black" cluster with intensely racist terms such as "shit", "fucking", and "stupid". Nowhere in these ATA analyses does "black" appear alongside gendered terms, which appear in nearly all other topics. Instead, terms such as "dude" cluster with "black" as a way to simultaneously lock Black transgender girls out of gender on two fronts: they are neither boys nor girls. They are simply interlopers indexed as "dudes".

## 7. Conclusions

This study of 60,000 comments made on YouTube videos discussing transgender athletes found that these discussions cluster into three dominant themes: biology as determinative of athletic ability, racialized understandings of who constitutes a proper girl, and sex segregation in sports as the only way to ensure fairness, which is understood as winning.

While it may be tempting to look at these findings and conclude that they point to the need to desegregate sports entirely, research on the politics of gender identity and sexuality indicate that harnessing strategies for rights gains that Alison Gash (2015) terms "below the radar"—or doing so in such a way as to not draw opposition—might also be useful tactics to employ. These steps might look something like desegregating youth sports at very young ages where there is no rational purpose for separating competition by sex (Davis 2017), relying on try outs to determine level of competition in speeds that require strength and speed (Sharrow 2021), and revisiting sex-segregated sports, such as golf, shooting, distance swimming and running, where there are very few differences between men and women in outcomes (Leong 2017).

Additionally, these findings reveal that people receiving their news from YouTube and engaging in dialogue in the comments sections have a limited understanding of gender, biology, and transgender identification. This lack of information is likely shaped by prevailing patriarchal and heteronormative ideologies. Interest groups and social movements aiming to combat this disinformation should take steps to combat these fallacies by promoting awareness of transgender and gender issues aimed at educating the broader public. Elite messaging may complement these efforts, especially from non-partisan sources such as celebrities and social media influencers. Across the board, special care should be taken to highlight the ways that white supremacy shapes attitudes about binary gender and who, exactly, qualifies as a proper girl or boy.

This study also underscores the utility of descriptive studies in the social sciences by providing insight into how people talk about issues on social media. As venues such as YouTube, Facebook, Twitter, and Instagram increasingly take the place of traditional news sources, methods such as GTM analyses of comments paired with ATA stand to offer significant leverage on questions of how and why certain political and social problems—such as marginalization and stigmatization of groups—develop, endure, and morph over time.

**Funding:** This study received no external funding.

**Institutional Review Board Statement:** Not applicable.

**Informed Consent Statement:** Not applicable.

**Data Availability Statement:** The data in this study can be made available upon reasonable request to the corresponding author.

**Conflicts of Interest:** The author declares no conflict of interest.

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
