# Peer review of "Don’t Read the Comments: Examining Social Media Discourse on Trans Athletes"

_laws_

Round 1
Reviewer 1 Report
Thank you for the opportunity to review this manuscript on the social media discourse on trans athletes. It provides an interesting insight into public discourse on an evolving area of gendered political conversation. I have reflections and suggestions for strengthening the presentation of data and analysis.
First, on the description of methodological selection. Are there other peer-reviewed studies that analyze YouTube videos in the way of this study, and can the author engage with how those studies discuss both findings and limitations of this medium for data collection? Since the author makes analogous comparisons to studying YouTube for insight on public sentiment, how does this method relate to, say, the study of public opinion?
Second, on the case selection. Why these three videos? What kind of work was done to select them and what do we know about how representative they are of content addressing these issues on YouTube? Again, what does the literature that uses these methods suggest about selecting particular videos, and how did the author ground their selections in the extant research?
Third, on method. Can the author address some confusion as to the first steps of processing the data (i.e. comments)? Were the top themes/keywords presented entirely machine-generated? Or did the researcher go in with prior expectations on the content they would find with respect to the themes (and were these grounded in the literature on bill content, or in something else)? This is unclear in the submitted manuscript.
Fourth, on interpreting the data. The qualitative analyses are strong but I think readers could use more guidance in how to understand the content presented in the Tables. Anyone unfamiliar with ATA will struggle to understand how these tables were constructed, and what information is meant to be gleaned from them.
Fifth, the setup on race and gendered constructions is strong, but can the author make clear how racialized comments manifest directly (if at all) in comments? It’s unclear if the racialized commentary is directly inferred from comments, or if the author is providing a racialized analysis on the comments as given. Here again, if there are other studies published that use YouTube to analyze racialized discourse it could be helpful.
Finally, on the question of future directions/takeaways for policy/discursive outcomes, the paper could benefit from engaging with and citing a couple of additional recent pieces (specifically in the section on “Fairness and Testosterone”):
Sharrow, Elizabeth. 2021. “Sex Segregation as Policy Problem: A Gendered Policy Paradox.” Politics, Groups, and Identities 9(2): 258–79.
Currah, Paisley. 2022. Sex Is as Sex Does: Governing Transgender Identity. New York: New York University Press.
I appreciated the opportunity to review this manuscript and look forward to watching it develop.
Author Response
I would like to thank the reviewer for the thoughtful and constructive feedback on “Biology and Fairness: Examining Social Media Discourse on Trans Athletes.” I have revised the manuscript with their comments in mind, and detail the steps taken below.
First, on the description of methodological selection. Are there other peer-reviewed studies that analyze YouTube videos in the way of this study, and can the author engage with how those studies discuss both findings and limitations of this medium for data collection? Since the author makes analogous comparisons to studying YouTube for insight on public sentiment, how does this method relate to, say, the study of public opinion?
Thank you for this suggestion. I have revised section 3, Data and Methods, with these comments about the limitations and generalizability of YouTube data, taking care to explain that it is not a substitute for public opinion data precisely because it is not a random sample. An article by Bessi et al (2016) helps me underscore the reasons I use YouTube, specifically the insight that it provides in how conversations take shape in online social media forums and the degree to which it is generalizable. Bessi and co-authors find that YouTube users tend to correspond to Facebook users, and I use these findings to explain why I use YouTube data and why I expect it matters. I also elaborated each of the six reasons I list for using YouTube data.
Second, on the case selection. Why these three videos? What kind of work was done to select them and what do we know about how representative they are of content addressing these issues on YouTube? Again, what does the literature that uses these methods suggest about selecting particular videos, and how did the author ground their selections in the extant research?
I elaborate why I select these three videos. In brief, I use prior knowledge about the publishers as well as the tone to classify the videos as neutral, positive, or negative. It is impossible to assess the impact or representativeness of YouTube videos, which I address in the revised manuscript and explain how I use number of views – millions or tens of thousands – to motivate case selection. I also provide information on key words I used for my searches.
Third, on method. Can the author address some confusion as to the first steps of processing the data (i.e. comments)? Were the top themes/keywords presented entirely machine-generated? Or did the researcher go in with prior expectations on the content they would find with respect to the themes (and were these grounded in the literature on bill content, or in something else)? This is unclear in the submitted manuscript.
Thank you for this question. I fleshed out the steps for the analysis so it would read more clearly. I also explained how iterative coding works for identifying repeated themes in the grounded coding.
Fourth, on interpreting the data. The qualitative analyses are strong but I think readers could use more guidance in how to understand the content presented in the Tables. Anyone unfamiliar with ATA will struggle to understand how these tables were constructed, and what information is meant to be gleaned from them.
I added information on how ATA works (somewhat like factor analysis) and how one interprets the results to the revised manuscript. The assumptions that undergird ATA are added along with relevant cites. I hope this helps to clarify the findings and how to read them.
Fifth, the setup on race and gendered constructions is strong, but can the author make clear how racialized comments manifest directly (if at all) in comments? It’s unclear if the racialized commentary is directly inferred from comments, or if the author is providing a racialized analysis on the comments as given. Here again, if there are other studies published that use YouTube to analyze racialized discourse it could be helpful.
I appreciate this comment. I went back through the analysis with an eye for drawing a thread throughout the analysis to underscore the ways that race and gender are intertwined in these comments.
Finally, on the question of future directions/takeaways for policy/discursive outcomes, the paper could benefit from engaging with and citing a couple of additional recent pieces (specifically in the section on “Fairness and Testosterone”):
Thank you for these suggestions! I added the Sharrow cite to the conclusion as that work helps to ground some of the possibilities I hint at in the conclusion. I also revised this section to explain that the recommendations offered are not to the exclusion of desegregating sports. This put the suggestions in conversation with scholarship as opposed to speculation.
Reviewer 2 Report
It is a great that that needs to be published! It makes a relevant contribution to the field of trans studies but most importantly, to the overall debate that is taking place nowadays.
Author Response
Thank you!
Reviewer 3 Report
Overall, this paper demonstrates a novel methodological approach to studying an important topic in public discourse. The author effectively frames the issue and establishes its broader importance, and the findings are clearly explained and discussed. Would recommend for publication with some fairly minor changes.
Introduction
· Compelling opening w/inclusion of a specific representative example/anecdote, followed by discussing the broader scope of the issue (i.e., the recent influx of anti-trans legislation).
· I appreciate that the author takes care to situate the rhetoric about trans athletes as being part of a much wider context of anti-trans public discourse, which does a good job of establishing why the issue is important.
Background and Theoretical Frame
· Author provides a useful explanation of how anti-trans rhetoric & legislation are linked to wider conservative frameworks about racial purity, work, biology, and citizenship.
· Other recent citations/literature that may be worth mentioning in this section:
o Erikainen, S. (2019). Gender verification and the making of the female body in sport: A history of the present. Routledge.
o Ivy, V., & Conrad, A. (2018). Including Trans Women Athletes in Competitive Sport: Analyzing the Science, Law, and Principles and Policies of Fairness in Competition. philosophical topics, 46(2), 103-140.
Data and Method
· It may help to expand the opening paragraph a bit, to give a bit more context on the analysis of YouTube comments as research data – is this a widespread/established practice? Any citations that could be added?
· In same paragraph, “a growing number of adults in the US report getting their news primarily from YouTube” – can you be more specific? How many adults, and/or can you state how dramatic/pronounced the increase has been?
· I would suggest explaining the ATA method in a little more detail, in terms of what constitutes a “topic” and how these categories are generated.
· The use of two methods simultaneously (grounded theory & automated text analysis) adds methodological strength, especially given that the two methods achieve similar themes/results.
Findings: Grounded Approach
· Very thorough and effective explanation of selected comments, particularly with regard to the racism and the dehumanization of the rhetoric around trans athletes.
· Since the author selected three videos with differing viewpoints – one neutral, one positive, and one negative with regard to trans inclusion in sports – I was surprised not to see more discussion of whether (if at all) the themes in the video comments differed based on which video was being responded to. This is alluded to briefly (i.e., stating that many comments responding to the ACLU video on the supportive father celebrated his love and acceptance of her). One might expect an “echo chamber” effect where commenters primarily agree with the sentiment expressed in the video, a backlash effect where commenters leave negative or angry comments on videos they disagree with, or a mix of both. Was this something that was explored at all?
Findings: Automated Text Analysis
· This section might benefit from a little bit more explication of the tables, for readers who may be less familiar with ATA as a method – mostly in terms of how the words grouped within a topic are linked and what they imply when taken together.
Discussion
· The author’s attempts at linking the findings produced by both types of analyses are novel and interesting.
· Excellent choice to reiterate that transphobic framings are harmful to cisgender girls as well by reinforcing their position as naturally subordinate to boys.
· This is a minor wording quibble, but I am not sure that a general audience’s lack of understanding of testosterone science can be laid at the feet of “transgender advocates’” lack of success in educating the general public – that lack of understanding may speak more to the broader contexts of transphobia, rigid understandings of gender, and attachment to existing social hierarchies described in the piece’s introduction.
Conclusion
· There are some interesting suggestions made here, but I would be careful not to overstate the conclusions. For example, the statement “While it may be tempting to look at these findings and conclude that they point to the need to desegregate sports entirely, a far more effective approach that would benefit all transgender and gender non-conforming people might be to harness strategies for rights gains that Alison Gash (2015) terms “below the radar” – or doing so in such a way as to not draw opposition.” I am not sure that the findings from this paper conclusively suggest that one of those two approaches is “far more effective” than the other; it comes off more as the author’s personal preference/opinion, since this analysis does not compare the effectiveness of competing messaging strategies.
Author Response
I would like to thank the reviewers for their thoughtful and constructive feedback on “Biology and Fairness: Examining Social Media Discourse on Trans Athletes.” I have revised the manuscript with their comments in mind, and detail the steps taken below.
- It may help to expand the opening paragraph a bit, to give a bit more context on the analysis of YouTube comments as research data – is this a widespread/established practice? Any citations that could be added?
I elaborate why I select these three videos. In brief, I use prior knowledge about the publishers as well as the tone to classify the videos as neutral, positive, or negative. It is impossible to assess the impact or representativeness of YouTube videos, which I address in the revised manuscript and explain how I use number of views – millions or tens of thousands – to motivate case selection. I also provide information on key words I used for my searches.
In same paragraph, “a growing number of adults in the US report getting their news primarily from YouTube” – can you be more specific? How many adults, and/or can you state how dramatic/pronounced the increase has been?
I only have Pew data, which isn’t disaggregated by demographics. I added a discussion about how and why I expect this information to be useful in the context of this paper. In particular, the findings from Bessi et al 2016 shows that YouTube users tend to behave in similar ways to Facebook users. This suggests that examining YouTube comments provides a (limited) view in how people engage in dialogue on social media. I have clarified the scope of generalizability throughout the manuscript.
- I would suggest explaining the ATA method in a little more detail, in terms of what constitutes a “topic” and how these categories are generated.
I made significant revisions and additions to provide readers with information on how ATA works (somewhat like factor analysis) and how one interprets the results to the revised manuscript. I hope this helps to clarify the findings and explain how topics are generated.
Since the author selected three videos with differing viewpoints – one neutral, one positive, and one negative with regard to trans inclusion in sports – I was surprised not to see more discussion of whether (if at all) the themes in the video comments differed based on which video was being responded to. This is alluded to briefly (i.e., stating that many comments responding to the ACLU video on the supportive father celebrated his love and acceptance of her). One might expect an “echo chamber” effect where commenters primarily agree with the sentiment expressed in the video, a backlash effect where commenters leave negative or angry comments on videos they disagree with, or a mix of both. Was this something that was explored at all?
I appreciate this question and comment. I went back through the findings from the grounded analysis with these points in mind, and I was able to see that the comments were similar regardless of the video’s tone/ideological position. I explain in the paper at the consistency in comments suggests that there consistent ways that transgender people and athletes are understood. For that reason, I decided to maintain the structure of the section – by theme – rather than by video.
This section might benefit from a little bit more explication of the tables, for readers who may be less familiar with ATA as a method – mostly in terms of how the words grouped within a topic are linked and what they imply when taken together.
Thank you -- reviewer 1 also asked for this revision. I added information in section 5 on how to interpret ATA findings and used an analogy to a related method – factor analysis – to explain the assumptions that undergird the findings. I hope that these will help readers to understand the method, how it works, and how to interpret findings.
- This is a minor wording quibble, but I am not sure that a general audience’s lack of understanding of testosterone science can be laid at the feet of “transgender advocates’” lack of success in educating the general public – that lack of understanding may speak more to the broader contexts of transphobia, rigid understandings of gender, and attachment to existing social hierarchies described in the piece’s introduction.
Thank you for this point! I revised the concluding thoughts to underscore that these trends are the result of a more pervasive climate of transphobia, white supremacy, and patriarchy. I also added a paragraph at the end of section 6 to draw attention back to the racial dynamics at play, which dropped out in the previous version.
There are some interesting suggestions made here, but I would be careful not to overstate the conclusions. For example, the statement “While it may be tempting to look at these findings and conclude that they point to the need to desegregate sports entirely, a far more effective approach that would benefit all transgender and gender non-conforming people might be to harness strategies for rights gains that Alison Gash (2015) terms “below the radar” – or doing so in such a way as to not draw opposition.” I am not sure that the findings from this paper conclusively suggest that one of those two approaches is “far more effective” than the other; it comes off more as the author’s personal preference/opinion, since this analysis does not compare the effectiveness of competing messaging strategies.
- I appreciate this comment. I revised the conclusion to include a cite to Sharrow (2021) to ground these suggestions in the scholarship on trans issues/politics, and explain that these strategies work alongside broader efforts to desegregate certain sports/public spaces, not as a substitute.